# Temporal Preference Optimization of Large Multimodal Models

## Abstract

Despite recent advancements in video large multimodal models (video-LMMs), accurate temporal grounding remains a key challenge. In this work, we introduce **Temporal Preference Optimization** (**TPO**)—a post-training framework that unlocks superior temporal reasoning in video-LMMs without requiring human annotations. TPO enables preference modeling by manipulating video inputs to generate contrastive responses, ensuring that preferred responses are more temporally grounded than dis-preferred ones. Through preference learning, TPO enhances the model's capability for more comprehensive video understanding with better temporal reasoning. Extensive experiments on LongVideoBench, MLVU, and Video-MME demonstrate that TPO significantly improves temporal grounding across multiple video-LMMs. Notably, LLaVA-Video-TPO achieves state-of-the-art performance among 7B models on Video-MME, establishing TPO as a scalable and effective solution for advancing temporal understanding in video analysis.

## 1 Introduction

Recent advances in video large multimodal models (video-LMMs) (Wang et al., 2024b; Achiam et al., 2023; Reid et al., 2024) represents a significant step toward general video understanding. While image-based LMMs (Hong et al., 2024; Bai et al., 2023; Lu et al., 2024) primarily focus on spatial reasoning, video-LMMs face the additional complexity of modeling temporal dependencies—a critical aspect for capturing the dynamic nature of video content.

Most existing video-LMMs are trained through supervised finetuning with video–question–answer pairs, without explicit mechanisms for temporal grounding. Consequently, temporal alignment is only acquired implicitly, and models often struggle to localize the precise moments that support their responses (Chen et al., 2024a; Zhang et al., 2024d). Recent efforts (Ren et al., 2024; Chen et al., 2024b; Li et al., 2023a; Huang et al., 2024; Wang et al., 2024a) have sought to improve grounding by enriching textual responses with structured temporal information and incorporating explicit segment-level annotations into training. While providing stronger supervision, it relies on additional temporal annotations, which are costly to obtain and difficult to scale to large datasets.

We introduce Temporal Preference Optimization (TPO), a post-training framework that enhances temporal grounding in video-LMMs without requiring manual annotations. TPO generates contrastive supervision by prompting a model with the same query on both original and corrupted videos: responses from relevant frames are treated as preferred, while those from irrelevant or incomplete frames are dis-preferred, forming a natural preference hierarchy. A lightweight post-filtering step removes noisy or ambiguous samples, yielding a clean preference dataset. This dataset is then used to refine temporal grounding through Direct Preference Optimization (Rafailov et al., 2024), which improves temporal reasoning while preserving pretrained knowledge. By automatically injecting temporal preferences through simple input transformations, TPO provides a scalable and robust solution for advancing temporal reasoning tasks.

We conducted extensive experiments on three challenging multimodal video understanding benchmarks, and the results clearly demonstrate that TPO significantly enhances the temporal grounding capabilities of video-LMMs. Specifically, TPO achieves performance gains of 2.9% on LongVideoBench (Wu et al.,

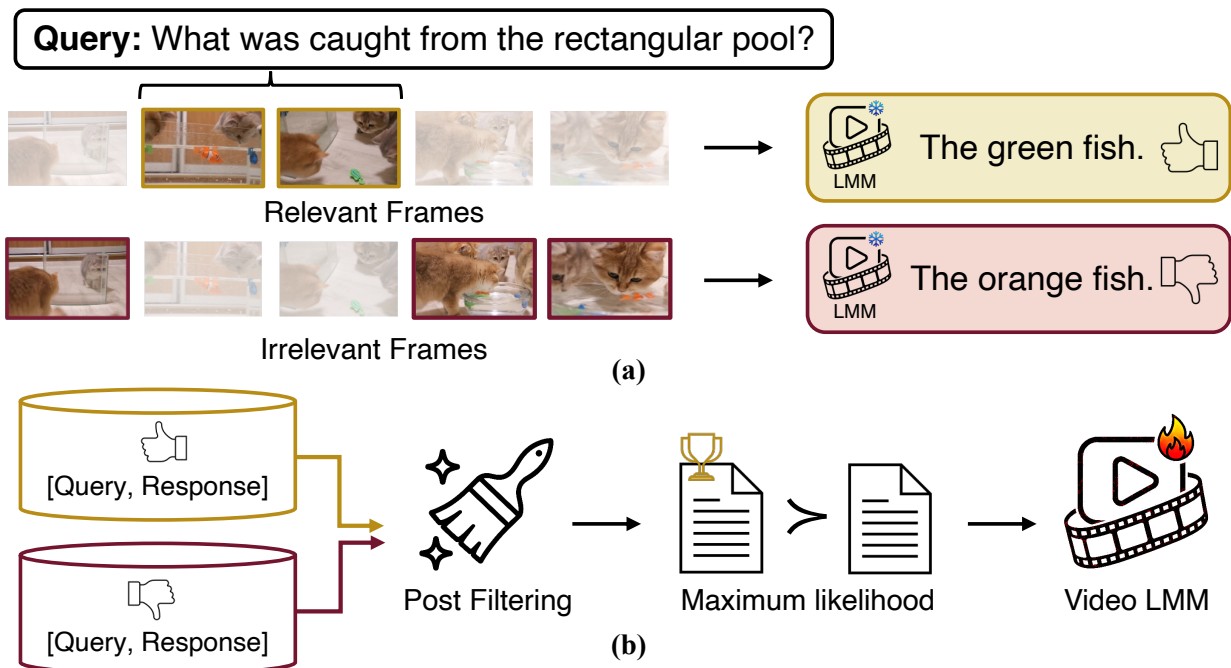

Figure 1: **Temporal Preference Optimization.** (a) The model self-generates preference data by producing contrastive responses to well-grounded versus perturbed (irrelevant or incomplete) video clips. (b) An LLM-based post-filtering step removes noisy or misaligned samples, and the refined preference data is then used in preference optimization. Through this self-improvement process, the model learns to favor temporally consistent responses, ultimately strengthening temporal reasoning.

2024), 3.1% on MLVU (Zhou et al., 2024a), and 2.5% on Video-MME (Fu et al., 2024), when applied to the strong base model LongVA-7B (Zhang et al., 2024b). Furthermore, even when integrated with the state-of-the-art large-scale pretrained video-LMM, LLaVA-Video, TPO still delivers a 2.3% improvement, establishing LLaVA-Video-TPO as the top-performing 7B model on the Video-MME benchmark.

## 2 Temporal Preference Optimization

While prior works focus primarily on aligning LLM outputs with human preferences, our approach uniquely aligns model outputs with intrinsic temporal preferences in videos. To achieve this, we propose Temporal Preference Optimization (TPO) (Fig. 2), a framework that enhances video-LMMs' temporal reasoning by explicitly incorporating temporal modeling into the optimization process. TPO generates preference pairs through contrasts between meticulously manipulated video inputs (Sec. 2.1). To further enhance the preference data quality, we introduce a rule-based post-filtering step (Sec. 2.2). Finally, Direct Preference Optimization (Sec. 2.3) is leveraged to optimize the model towards temporally preferred outputs without compromising its original pretrained capabilities.

### 2.1 Temporal Preference Modeling

**Query Generation.** Given a video $\mathbf{V}$, we first sample a segment containing a set of frames $S_a$, which may be a subset of the video or the entire sequence of frames. To generate descriptive context, we employ CogVLM2 (Hong et al., 2024), an image-based LMM, to generate captions for each frame in $S_a$. These captions serve as the foundation for constructing targeted questions. To ensure diversity and relevance, we design multiple question types and use a structured question-generation prompt to incorporate the generated captions, as shown in Fig. 9 (Appendix). This prompt is then processed by a LLM (GPT-4o-mini) to produce

a set of candidate questions specifically tailored to the sampled video frames, resulting in a set of questions $S_q$. This approach ensures that the generated questions are contextually relevant that allows precise control over subsequent response generation.

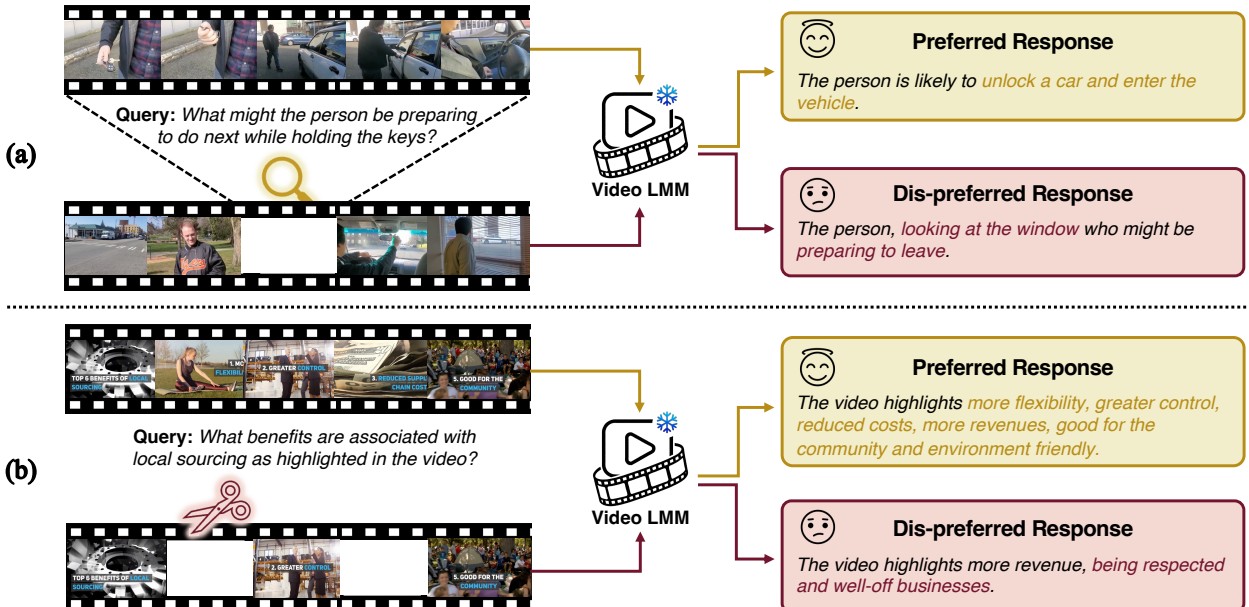

Figure 2: To ensure strong temporal grounding, we generate preferred responses using the full relevant frame set $S_a$. For dis-preferred responses, we introduce: (a) **Generation with Irrelevant Information**, where all relevant frames are excluded. (b) **Generation with Incomplete Information**, where only a partial subset of relevant frames is used. These manipulated clips create contrastive response pairs, highlighting differences between well-grounded and manipulated video content. This contrast serves as a learning signal to enhance the model's temporal reasoning.

**Preferred Response Generation.**   Preferred responses in the curated dataset are expected to be strongly grounded in the corresponding temporal content. To achieve this, we use the question set $S_q$ along with their corresponding frame set $S_a$ as input to the video-LMM. Given the provided video frames are highly relevant to the query, we create conditions that maximize the likelihood of generating a high-quality, temporally grounded response. This process guarantees that the preferred responses align with the ideal characteristics for effective temporal grounding in video-LMMs.

**Dis-Preferred Response Generation.**   In our preference dataset, dis-preferred responses are those that the model is trained to avoid—outputs that fail to temporally localize the evidence in the video. These serve as hard negative examples for temporal reasoning, highlighting cases where the model struggles to align its predictions with the actual video content. To generate these dis-preferred responses, we manipulate the video inputs to simulate imperfect temporal grounding. As illustrated in Fig. 2, we introduce two strategies for constructing the input frame set $S_b$ used in dis-preferred response generation:

**(a) Generation with Irrelevant Information**: To simulate an extreme failure case where the model misses all relevant frames, we construct $S_b$ by excluding the relevant frame set $S_a$ and instead sampling from the remaining frames of the video. This ensures that $S_b$ contains only irrelevant content, forcing the model to generate a response based on unrelated visual information.

**(b) Generation with Incomplete Information**: To simulate that the model can only consume partial relevant information, $S_b$ is randomly sampled as a subset of $S_a$. This setup introduces gaps in the temporal context, making it harder for the model to fully comprehend the key event in the query.

Unlike preferred responses, which are grounded in fully relevant video segments, these manipulated setups introduce ambiguity and noise by partially or completely omitting critical visual content. As a result, the model is forced to rely on incomplete or misleading information, making temporal reasoning errors and hallucinations more likely. This intentional contrast between preferred and dis-preferred responses serves as a strong learning signal, helping refine the model's ability to distinguish and accurately localize events in time, ultimately enhancing its temporal reasoning capabilities.

## 2.2 LLM-based Post-Filtering

Although we design the preferred responses to be higher quality than the dis-preferred responses, this distinction is not always guaranteed due to the limitations of the base video-LMMs. In some cases, errors in response generation may lead to misaligned preference pairs, where the preferred response contains noise or the dis-preferred response is of better quality than expected.

To enhance data quality and reduce noise, we introduce a post-filtering pipeline with an LLM (GPT-4o-mini). Specifically, we provide the model with the key frame captions of $S_a$, along with their corresponding queries and preference data pairs, and instruct it to filter out samples that meet predefined criteria (detailed prompts are shown in Fig. 10 in the Appendix). The filtering rules target cases where: 1) The dis-preferred response is of higher quality than the preferred response. 2) The preferred response is factually incorrect or misaligned with the video content. 3) The query is ambiguous, making preference ranking unreliable. By incorporating this post-filtering step, we effectively eliminate potential noisy cases, resulting in a refined, higher-quality dataset that better supports effective model optimization and improves temporal grounding performance.

## 2.3 Training Objective

The generated preference dataset is leveraged to optimize the temporal grounding capabilities of video-LMMs using Direct Preference Optimization (DPO) (Rafailov et al., 2024), selected for its robustness and effectiveness in preference-based learning.

Given the preference dataset D $(V, q, r^+, r^-)$ and a video-LMM $\pi_\theta$, the DPO loss is defined as:

$$L_{DPO}(\pi_\theta; \pi_{ref}) = -\mathbb{E}_{(V,q,r^+,r^-)\sim\mathcal{D}} \left[ \log \sigma \left( \beta \left( \log \frac{\pi_\theta(r^+|V,q)}{\pi_{ref}(r^+|V,q)} - \log \frac{\pi_\theta(r^-|V,q)}{\pi_{ref}(r^-|V,q)} \right) \right) \right] \tag{1}$$

where $\sigma$ is the sigmoid function. This objective drives the model to assign higher probabilities to preferred outputs, aligning its behavior more closely with human judgments, while preventing the model from deviating too much from its pretrained distribution.

To better align the model with the preferred responses, we incorporate a supervised fine-tuning objective into the DPO training framework. This combined objective is controlled by the hyperparameter $\alpha$, following (Chen et al., 2021; Deng et al., 2024; Chen et al., 2023).

$$L_{SFT}(\pi_\theta) = -E_{(V,q,r^+,r^-)\sim\mathcal{D}} \log \pi_\theta(r^+|V,q) \tag{2}$$

$$L(\pi_\theta; \pi_{ref}) = L_{DPO} + \alpha L_{SFT} \tag{3}$$

# 3 Experiments

## 3.1 Experimental Settings

**Evaluation Benchmarks** We evaluate TPO and baselines on three widely recognized benchmarks in multimodal video understanding.

- **Video-MME** (Fu et al., 2024) offers a comprehensive multi-modal evaluation across diverse video lengths, spanning from 11 seconds to 1 hour.

| Model | Size | LongVideo Bench | MLVU (M-avg) | Video-MME | | | |
| --- | --- | --- | --- | --- | --- | --- | --- |
| | | | | Short | Medium | Long | Average |
| GPT-4o | - | 66.7 | 64.6 | 80.0/82.8 | 70.3/76.6 | 65.3/72.1 | 71.9/77.2 |
| Video-LLaVA | 7B | 39.1 | 47.3 | 45.3/46.1 | 38.0/40.7 | 36.2/38.1 | 39.9/41.6 |
| PLLaVA | 7B | 40.2 | - | - | - | - | - |
| Qwen-VL-Max | - | - | 42.2 | 55.8/57.6 | 49.2/48.9 | 48.9/47.0 | 51.3/51.2 |
| ShareGPT4Video | 8B | 39.7 | 46.4 | 48.3/53.6 | 36.3/39.3 | 35.0/37.9 | 39.9/43.6 |
| InternVL-Chat-V1.5 | 20B | 51.2 | 50.4 | 50.7/52.4 | 60.2/61.7 | 46.4/49.1 | 45.6/46.6 |
| VideoChat2 | 7B | 39.3 | 47.9 | 48.3/52.8 | 37.0/39.4 | 33.2/39.2 | 39.5/43.8 |
| LongLLaVA | 7B | - | 56.3 | 61.9/66.2 | 51.4/54.7 | 45.4/50.3 | 52.9/57.1 |
| Video-CCAM | 14B | - | 63.1 | 62.2/66.0 | 50.6/56.3 | 46.7/49.9 | 53.2/57.4 |
| NVILA | 7B | 57.7 | 70.1 | 75.7/77.6 | 62.2/69.0 | 54.8/63.3 | 64.2/70.0 |
| Qwen2-VL | 7B | 55.6 | - | - | - | - | 63.3/69.0 |
| Apollo | 7B | 58.5 | 70.9 | - | - | - | 61.3/63.3 |
| MiniCPM-o-2.6 | 7B | - | - | 75.4/78.3 | 63.9/69.1 | 52.2/56.3 | 63.9/67.9 |
| LongVILA | 7B | 57.1 | - | 69.0/72.9 | 58.3/64.9 | 53.0/57.4 | 60.1/65.1 |
| LiveCC | 7B | - | - | 74.8/76.6 | 63.9/70.3 | 53.7/64.1 | 64.1/70.3 |
| Qwen2.5-VL | 7B | 56.0 | 70.2 | - | - | - | 65.1/**71.6** |
| LongVA-7B | 7B | 51.3 | 58.8 | 61.1/61.6 | 50.4/53.6 | 46.2/47.6 | 52.6/54.3 |
| LLaVA-Video-7B | 7B | 58.2 | 70.8 | - | - | - | 63.3/69.7 |
| **LongVA-TPO** | 7B | 54.2 | 61.7 | 63.1/66.6 | 54.8/55.3 | 47.4/47.9 | 55.1/56.6 |
| **LLaVA-Video-TPO** | 7B | **60.1** | **71.1** | **76.8/78.7** | **64.6/69.4** | **55.4/66.4** | **65.6**/71.5 |

Table 1: Results on LongVideoBench (Wu et al., 2024), MLVU (Zhou et al., 2024a) and Video-MME (Fu et al., 2024) compared with state-of-the-art models. The Video-MME results are presented in the format "w/o subs / w/ subs".

- **LongVideoBench** (Wu et al., 2024) emphasizes reasoning tasks within extended video contexts.

- **MLVU** (Zhou et al., 2024a) supports multitask evaluation specifically designed for long-form video understanding.

**Models**   We test the effectiveness of TPO on two popular video-LMMs, LongVA-7B (Zhang et al., 2024b) and LLaVA-Video-7B (Zhang et al., 2024e), deriving the following models:

- **LongVA-TPO**: optimized based on LongVA-7B (Zhang et al., 2024b), a capable video-LMM with the long-context video understanding capability transferred from language.

- **LLaVA-Video-TPO**: optimized based on LLaVA-Video-7B (Zhang et al., 2024e), the current state-of-the-art 7B video-LMM.

In the main text, all ablation studies and analyses are conducted using LongVA-TPO by default; additional ablation results are provided in the appendix.

**Implementation Details**   For the video source in preference dataset generation, we manually curated 200 keywords to retrieve 8k videos from the internet, ensuring diversity and coverage. For each video, we sampled 32 frames. To simulate irrelevant information, we randomly selected 4 consecutive frames as $S_a$ and used the remaining 28 frames as $S_b$. To simulate incomplete information, we instead used all 32 frames as $S_a$ and uniformly removed 16 frames to form $S_b$. Using this pipeline, we created 10k preference data pairs for LongVA-TPO using our established pipeline. For LLaVA-Video-TPO, we employ a subset of the original LLaVA-Video-178K dataset, which was used for supervised fine-tuning (SFT), to generate TPO data, resulting in a total of 10K preference data pairs.

| Model | LongVideo Bench | MLVU (M-avg) | Video-MME | | | |
|---|---|---|---|---|---|---|
| | | | **Short** | **Medium** | **Long** | **Average** |
| LongVA-7B | 51.3 | 58.8 | 61.1/61.6 | 50.4/53.6 | 46.2/47.6 | 52.6/54.3 |
| + $SFT_{Self}$ | 52.7 | 58.9 | 62.6/**67.7** | 52.4/52.7 | 46.8/47.4 | 53.9/55.9 |
| + $SFT_{LLM}$ | 53.1 | 59.9 | **63.7**/64.9 | 52.6/54.3 | 46.3/47.9 | 54.2/55.7 |
| + Hound-DPO[†] | 52.8 | 59.1 | 62.2/65.8 | 52.4/54.8 | 46.1/46.3 | 53.6/55.6 |
| + Hound-DPO[*] | 52.6 | 59.3 | 63.1/65.9 | 50.8/54.7 | 47.2/47.0 | 53.7/55.9 |
| **LongVA-TPO** | **54.2** | **61.7** | 63.1/66.6 | **54.8/55.3** | **47.4/47.9** | **55.1/56.6** |

Table 2: Results of LongVA-TPO on LongVideoBench (Wu et al., 2024), MLVU (Zhou et al., 2024a) and Video-MME (Fu et al., 2024) benchmarks compared to baseline methods mentioned in 3.2. The Video-MME results are presented in the format "w/o subs / w/ subs". The results for LongVA and LongVA+Hound-DPO[†] (Zhang et al., 2024c;b) are based on publicly available checkpoints, while LongVA+Hound-DPO[*] is reproduced using our collected video datasets.

The model is trained on 8 Nvidia A100 80GB GPUs, with a batch size of 64. For the preference optimization on LongVA, we set the KL-divergence weight $\beta = 0.3$ and the SFT loss weight $\alpha = 0.5$, while for LLaVA-Video, we set the KL-divergence weight $\beta = 0.2$ and the SFT loss weight $\alpha = 1$. To ensure information consistency, we use the same sampled 32 frames for both data generation and model training. We train the model on our curated data for 1 epoch. It takes about 4 hours for TPO to perform on LongVA-7B with a learning rate of $4e^{-6}$ and also about 4 hours for LLaVA-Video-7B with a learning rate of $3e^{-7}$. During data preparation, we employ the GPT-4o-mini (text-only input) for question curation and post-filtering. This choice balances cost-effectiveness with efficiency, facilitating a streamlined and scalable data processing workflow.

## 3.2 Results

The comparisons between TPO and current state-of-the-art video-LMMs on LongVideoBench (Wu et al., 2024), MLVU (Zhou et al., 2024a), and Video-MME (Fu et al., 2024) are presented in Table 1. With the introduction of TPO, both the LongVA-TPO and LLaVA-Video-TPO models significantly outperform their corresponding baselines by 2.5% and 2.3% on the Video-MME benchmark, demonstrating the efficacy of our TPO pipeline. After TPO on LLaVA-Video-7B, our LLaVA-Video-TPO model outperforms all 18 baseline models in the table, including the concurrent work, as well as several 14B and 20B models, achieving state-of-the-art results on video understanding. The original LongVA model performed worse than Video-CCAM (Fei et al., 2024) and LongLLaVA (Yin Song and Chen Wu and Eden Duthie, 2024) on the Video-MME benchmark. However, after incorporating TPO, it successfully outperformed these competitive baselines on Video-MME. Overall, LLaVA-Video-TPO achieves the strongest 7B model on Video-MME, setting a new state-of-the-art performance on video comprehension.

Besides, we compare TPO against three different training strategies on LongVA in Table 2:

- $SFT_{Self}$: Supervised fine-tuning using the self-generated data. For a fair comparison, we utilize the same preferred response in our curated preference dataset to optimize LongVA.

- $SFT_{LLM}$: Supervised fine-tuning using the LLM-generated data. Following the commonly used data curation pipeline (Chen et al., 2024a; Zhang et al., 2024d). We employ LLM (GPT-4o-mini) to generate responses given the query and the video captions, which are subsequently used to perform supervised fine-tuning on LongVA. We use the same video data as TPO for fair comparison.

- Hound-DPO (Zhang et al., 2024c): Applying Direct Preference Optimization (DPO) (Rafailov et al., 2024) to video-LMMs, Hound-DPO leverages LLM to rate preference data, resulting in a dataset of 17k samples. In contrast, TPO relies on a smaller, self-generated preference dataset, offering a more streamlined alternative. Besides, to ablate the data source's effect, we also implement Hound-DPO based on our collected dataset with the same data scale.

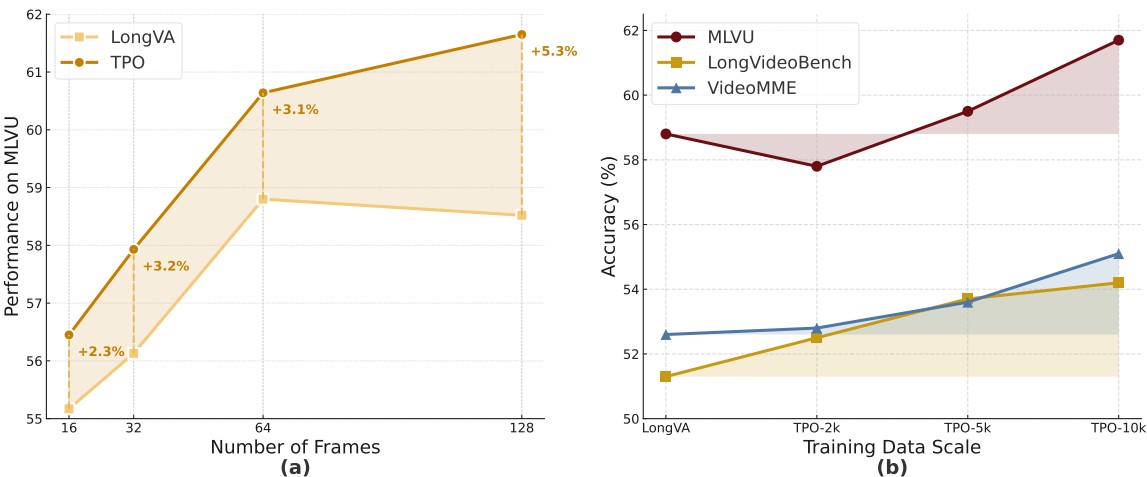

Figure 3: (a) Performance of LongVA-TPO vs. LongVA on MLVU across input lengths. LongVA-TPO consistently benefits from increased input length, whereas LongVA's performance declines when inputs exceed 64 frames. (b) The consistently improved performance of LongVA-TPO when scaling the data. The performance on the Video-MME benchmark is evaluated without subtitles.

The results consistently indicate that LongVA-TPO achieves superior performance, with improvements of 2.9%, 3.1%, and 2.5% on LongVideoBench (Wu et al., 2024), MLVU (Zhou et al., 2024a), and Video-MME, respectively. These findings underscore TPO's capacity to enhance the general video understanding capabilities of a pre-trained video-LMM.

Compared to SFT$_{\text{Self}}$, LongVA-TPO achieves a consistent performance gain of 1.2% to 2.8% by utilizing carefully designed temporal preference pairs. Furthermore, LongVA-TPO outperforms SFT$_{\text{LLM}}$, demonstrating the effectiveness and stability of our self-training paradigm. When compared to Hound-DPO (Zhang et al., 2024c), LongVA-TPO achieves a significant performance improvement by modeling temporal preference priors. However, LongVA-TPO achieves comparable performance compared to SFT methods on the Video-MME-short subset, which aligns with TPO's design focus—enhancing temporal reasoning in longer videos.

### 3.3 Analysis

**Effect of Input Frame Count**  We evaluate the performance of LongVA-TPO and LongVA across input lengths ranging from 16 to 128 frames, as shown in Fig. 3(a). While LongVA's performance declines when extending the input from 64 to 128 frames, LongVA-TPO continues to improve consistently with longer sequences. This trend indicates that TPO effectively leverages additional temporal context without suffering from information dilution or attention drift. The results highlight LongVA-TPO's robustness to extended inputs and its enhanced ability to retrieve and reason over relevant information across long temporal spans, underscoring the effectiveness of temporal preference optimization.

**Effect of Dataset Sizes**  Scalability is a critical metric in the evaluation of algorithms in the era of large-scale models, reflecting an algorithm's performance as data volume expands. To assess this, we evaluate LongVA-TPO on across incremental sizes of 2k, 5k, and 10k. As shown in Fig. 3(b), performance improves consistently with larger datasets across all three benchmarks, demonstrating superior scalability. This pattern highlights TPO's robustness and adaptability in larger data contexts, suggesting its potential to deliver enhanced results when scaled to larger datasets.

**Effect of Post-Filtering**  As a critical component of the TPO framework, post-filtering effectively reduces noise and enhances data quality. After post-filtering, we manually reviewed 200 preference pairs and found that 96.5% satisfied our criteria—where the preferred response was both more appropriate than the dis-preferred

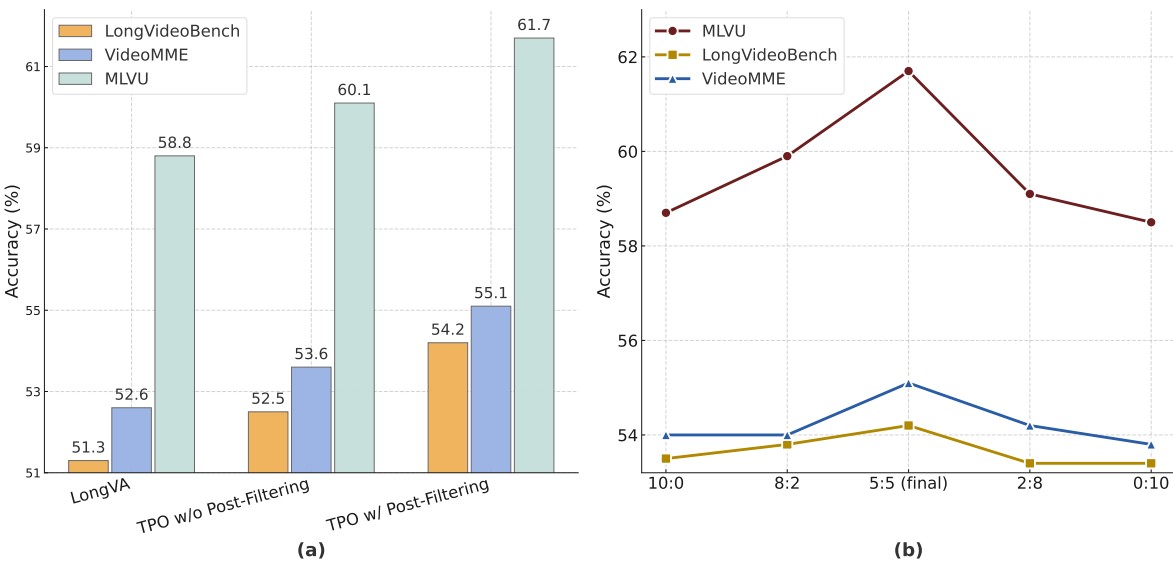

Figure 4: (a) Post-filtering consistently improves performance across multiple benchmarks on LongVA. (b) Performance of TPO across different training data mix ratios, varying the proportion of negative responses generated from incomplete versus irrelevant video segments.

one and accurately answered the question based on the video content. To further assess its impact, we conducted experiments comparing the performance of LongVA-TPO with and without post-filtering. The results, presented in Fig. 4 (a), demonstrate that post-filtering consistently improves performance across multiple benchmarks.

**Effect of Different Data Mix Ratio**   In TPO, we introduce two complementary manipulation schemes for constructing dis-preferred responses: one injects irrelevant temporal content, while the other introduces incomplete information. To understand their individual contributions and the benefit of combining them, we conduct an ablation study under a fixed total dataset size. Specifically, we vary the mixing ratios of the two schemes across 10:0, 8:2, 5:5, 2:8, 0:10.

As shown in Fig. 4(b), performance peaks at the 5:5 setting. This balanced mixture yields the best results, suggesting that integrating both types of temporal perturbations allows the model to learn more robust temporal discrimination. In particular, combining irrelevant and incomplete evidence helps TPO better suppress spurious temporal cues while reinforcing the model's ability to recognize missing or partially observable information, ultimately leading to stronger overall performance.

**Needle-in-a-Haystack**   The Needle-in-a-Haystack (NIAH) task challenges models to detect and reason about rare events within extremely long videos, where the signal of interest occupies only a tiny fraction of the total duration. Following Zhang et al. (2024b), we frame the task using the same five image-based question answerings (QAs), where images are embedded within a 3-hour-long video. The model must correctly identify the relevant temporal segment and answer the associated image QA, effectively testing both long-range temporal retrieval and fine-grained reasoning.

As shown in Fig. 5, while LongVA—optimized for efficient long-context modeling—already surpasses LLaVA-NeXT-Video (Zhang et al., 2024d) on the NIAH task (refer to Fig. 4 in Zhang et al. (2024b)), our LongVA-TPO model achieves further gains by explicitly optimizing temporal preference alignment. The improvement highlights TPO's ability to enhance temporal grounding over extended sequences.

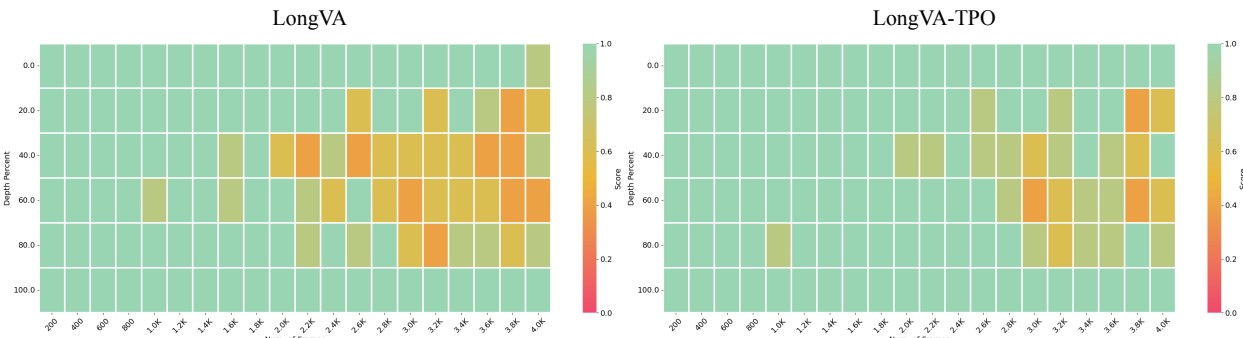

Figure 5: Performance comparison of LongVA and LongVA-TPO on the needle-in-a-haystack task across varying input video lengths (horizontal axis) and temporal depths (vertical axis). Heatmaps indicate improved temporal grounding capability of LongVA-TPO.

### 3.4 Qualitative Analysis

The qualitative analysis of LongVA-TPO and LongVA on two videos from the Video-MME benchmark is provided in Fig. 6. In the first example, which involves a temporal localization and OCR task, LongVA-TPO accurately localizes the relevant information within the video and providing the correct answer to the OCR question. In the second example, a video discussing the Moon's formation, LongVA misinterprets the video content by relating it to the Earth's formation. In contrast, LongVA-TPO successfully comprehends and captures the key details of the video's content.

## 4    Related Work

**Video Large Multimodal Models (video-LMMs)**    Recently, significant efforts have extended large language models (Achiam et al., 2023; Reid et al., 2024) into the visual domain, leading to the development of both proprietary (Achiam et al., 2023; Reid et al., 2024) and open-source video-LMMs (Wang et al., 2024b; Liu et al., 2024a; Li et al., 2024b; Shen et al., 2024; Lin et al., 2023; Chen et al., 2024c; Yao et al., 2024; Fu et al., 2025; Abdin et al., 2024; Laurençon et al., 2024; Chen et al., 2025d;a; Bai et al., 2025). Early work emphasized video-text instruction-tuning datasets (Chen et al., 2024a; Zhang et al., 2024e; Liu et al., 2023; Park et al., 2023; Zhang et al., 2024d), but their reliance on synthetic captions limits their effectiveness in capturing visual-temporal dynamics. Other studies have focused on extending pretrained video-LMMs for long contexts (Zhang et al., 2024b; Liu et al., 2024d; Yin Song and Chen Wu and Eden Duthie, 2024; Liu et al., 2024b;c; Shu et al., 2024; Islam et al., 2025), while multimodal interleaved datasets (Li et al., 2024c; Lin et al., 2024) and mixed training strategies (Zohar et al., 2024b; Li et al., 2024a) have been explored to enhance video-LMM performance. However, the post-training stage for video-LMMs remains underexplored. Recent efforts like LLaVA-Hound (Zhang et al., 2024c) builds preference datasets by ranking model outputs with LLM but fall short in leveraging the temporal information inherent in video. In contrast, our work pioneers post-training strategies that explicitly incorporate temporal priors.

Temporal grounding is crucial for comprehending the video modality, particularly in long-form videos. Prior work has explored diverse strategies, including dense captioning (Wang et al., 2021; Yeung et al., 2018; Yang et al., 2023), highlight detection (Lei et al., 2021; Moon et al., 2023), and temporal video grounding (Gao et al., 2017; Yuan et al., 2019; Xiao et al., 2024), among others. Recent advancements have introduced temporal-aware designs in video-LMMs (Ren et al., 2024; Chen et al., 2024b; Li et al., 2023a; Huang et al., 2024; Wang et al., 2024a) and agentic systems with temporal grounding capabilities have emerged (Wang et al., 2025a). Besides, reinforcement learning (Shao et al., 2024; Feng et al., 2025; Chen et al., 2025c; Ouyang et al., 2025; Yu et al., 2025) has been recently applied to temporal grounding (Chen et al., 2025b; Wang et al., 2025b;c; Li et al., 2025; Su et al., 2025). Despite providing explicit temporal supervision, these methods require additional temporal annotations, which are costly and often impractical at large scales.

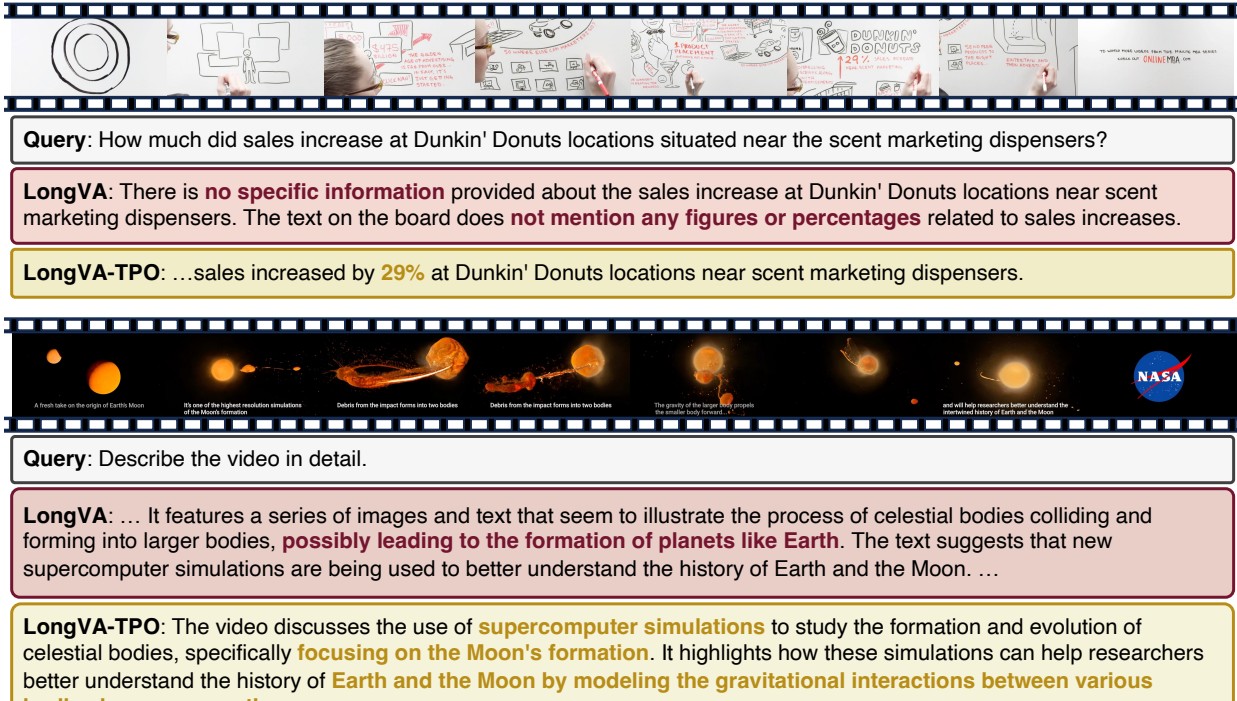

Figure 6: Qualitative comparison between LongVA-TPO and LongVA on videos from Video-MME.

Proximal Policy Optimization (Ouyang et al., 2022; Stiennon et al., 2020; Ziegler et al., 2019) and Direct Preference Optimization (Rafailov et al., 2024) are two widely used implementations of Reinforcement Learning from Human Feedback (RLHF) (Ouyang et al., 2022; Ziegler et al., 2019), serving as key algorithms in preference learning and post-training. In image-LMMs, Sun et al. (2023) improved model's visual capability by incorporating image captions into the reward model. Similarly, Ahn et al. (2024b) fine-tuned multimodal models using Reinforcement Learning from AI Feedback. Other approaches distilled GPT-4V's preferences directly from sampled responses (Li et al., 2023b; Gunjal et al., 2024) or leveraged text as an intermediate modality, using captions and other descriptive information to extract preferences for images (Zhao et al., 2023) and videos (Zhang et al., 2024c). Furthermore, Pi et al. (2024); Zhou et al. (2024b); Deng et al. (2024) advanced preference learning in image-LMMs by curating preference data through image input manipulation.

**Self-Training in Foundation Models**   To reduce reliance on large annotated datasets, several works have explored self-improvement and self-training methods (Huang et al., 2022; Ho et al., 2022). Zelikman et al. (2022) introduced Self-Taught Reasoners, which leverage generated chain-of-thought rationales to enhance LLMs' complex reasoning capabilities. For images, BPO (Pi et al., 2024), STIC (Deng et al., 2024) and POVID (Zhou et al., 2024b) improve image-LMMs responses by incorporating visual priors. For videos, Video-STaR (Zohar et al., 2024a) uses existing labels as weak supervision while Ahn et al. (2024a) explores iterative self-improvement in preference optimization.

## 5   Conclusion

We introduced Temporal Preference Optimization (TPO), a scalable post-training framework that enhances temporal grounding in video-LMMs. By contrasting between the preference responses from the well-grounded and manipulated video clips, TPO effectively captures the intricate temporal dependencies required for video understanding. Extensive experiments across three challenging benchmarks—LongVideoBench, MLVU, and Video-MME—demonstrated TPO's robust improvements, achieving state-of-the-art performance. By integrating multi-granularity temporal preferences, TPO offers a robust and efficient solution for advancing

temporal reasoning in multimodal tasks. One future direction is scaling the preference data to improve coverage and diversity, thereby enhancing TPO's generalizability. Additionally, while this work focuses on LongVA-7B and LLaVA-Video-7B as representative Video-LMMs, applying TPO to a broader range and larger scale of video-LMMs would provide insights into its adaptability and performance across different architectures.

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
