# OpenReview forum: "Temporal Preference Optimization of Large Multimodal Models"
_TMLR — Withdrawn by Authors_

### Review · Reviewer_tLjG · 2026-02-24

**Summary Of Contributions:**

This work targets the key limitation of video-LMMs in temporal reasoning, namely that predicted answers are often misaligned with the corresponding time segments. This work proposed a new post-training framework, termed Temporal Preference Optimization (TPO). The central claim is that TPO requires no additional human annotation, it automatically constructs preference pairs and perturbations, and applies post-filtering for data cleaning, enabling preference-based optimization of video large multimodal models. Extensive experiments across multiple benchmarks demonstrate the effectiveness of the proposed approach.

**Audience:**

Yes

**Audience Explanation:**

The proposed post-training paradigm—using preference learning for temporal alignment—appears reasonably extensible. It also targets a practical bottleneck in prior work, namely the high cost of manual annotation, which lends the approach applied value. However, the overall pipeline relies on GPT-4o-mini for question generation and filtering, which may hinder reproducibility. Overall, given the growing interest in video LLMs, I believe this work will attract the attention of a subset of TMLR's audience.

**Broader Impact Concerns:**

This work does not appear to raise any significant ethical concerns.

**Claims And Evidence:**

Yes

**Claims Explanation:**

Overall, several of the authors’ core claims are supported by empirical evidence to some extent. Regarding effectiveness, beyond reporting comparative results on multiple benchmarks, the paper further analyzes the impact of the number of sampled frames and video length, and provides ablation studies for individual components. However, given my limited familiarity with this specific field, I cannot confidently assess whether the comparison set includes the strongest and most widely used baselines, I will weigh this point together with feedback from other reviewers.

**Requested Changes:**

1. Post-filtering ablations. Consider a more fine-grained ablation of the post-filtering stage, e.g., how different filtering strengths and combinations of filtering rules affect performance, and whether using alternative open-source LLMs for post-filtering yields comparable gains.

2. Limitations and failure modes. The paper appears to lack an explicit limitations analysis. It would be valuable to document cases where TPO underperforms—such as temporal misalignment, over-generalization caused by missing evidence frames, or subtitle-driven hallucinations—to clarify the method’s boundary conditions and guide future improvements.

3. Reproducibility details. Please include the concrete post-filtering prompts/rules, key hyperparameters (e.g., sampling strategy and filtering thresholds), and the versioning and invocation settings of any external models in the main text or appendix to strengthen reproducibility and transferability.

---

### Review · Reviewer_ZEo7 · 2026-03-01

**Summary Of Contributions:**

### Summary

This paper proposes a data construction and post-training framework, termed Temporal Preference Optimization (TPO), to improve temporal grounding in video-language models. The key idea is to curate temporally aware preference data and apply a preference optimization strategy that encourages models to better align generated responses with temporally relevant video segments.


### Strengths
- The writing is quite fluent and easy to follow.
- The motivation of the paper is very clear.
- Experiments on LongVA and LLaVA-Video support that TPO improves temporal grounding-related performance compared to their original counterparts.

### Weaknesses (my concerns)
- The authors conducted experiments on two baselines, LongVA and LLaVA-Video, proving that their proposed TPO strategy is effective. However, looking at Table 1, models like NVILA, Apollo, and Qwen2.5-VL are clearly stronger. I have a question: why not experiment on these better baselines?
- In Figure 3, why does the performance on the MLVU dataset drop a bit at the TPO-2k scale?
- When testing on Needle-in-a-Haystack, I would like to know exactly how many frames were input into the model？
- Qwen-VL now has a 3.5 series. Is it possible to include a comparison with this?
- I noticed that the scores for Short Video, Medium Video, and Long Video are left blank for many of the baselines. Why didn't the authors report these?
- Out of curiosity, could the author run experiments on model sizes other than 7B?
- Regarding the evaluation metrics: since the authors evaluate using QA-based benchmarks, does correctly answering a content-related question really indicate better temporal grounding? I'm thinking about the rationality of this metric. Does good overall performance actually mean addressing the limitation mentioned at the beginning? Is there a way to decouple this? For example, there could be a situation where the model localizes the timestamp accurately but misunderstands the content. Is there a way to distinguish that scenario from a complete failure to localize?
- What does "temporal depths" mean? I didn't quite get it while reading, which made it hard for me to fully understand Figure 5.
- The reference policy in Eq. (1), π_ref, is used but not clearly defined in the text.

**Audience:**

Yes

**Audience Explanation:**

Improving temporal grounding and reasoning in large multimodal models is a highly active research area. Researchers working on video understanding, multimodal alignment, and preference optimization will find the proposed data construction and TPO pipeline relevant and interesting.

**Claims And Evidence:**

Yes

**Claims Explanation:**

I would say partially? While the authors provide clear evidence that TPO improves performance on their chosen baselines (LongVA and LLaVA-Video), the broader claims lack convincing support in two key areas. First, it is unclear if these improvements hold up against stronger, contemporary baselines like NVILA or Qwen2.5-VL. Second, the reliance on QA-based benchmarks does not convincingly isolate and prove improved "temporal grounding," as the metric conflates localization ability with general content understanding.

**Requested Changes:**

I have detailed my main questions and concerns in the Weaknesses section. To recommend acceptance, I primarily request that the authors address the clarity issues and experimental details.

---

### Review · Reviewer_Grqg · 2026-03-06

**Summary Of Contributions:**

This paper introduces Temporal Preference Optimization (TPO), a post-training framework designed to improve temporal grounding in Video Large Multimodal Models (video-LMMs) without manual annotations. The method generates preference pairs by contrasting responses from original video segments with those from corrupted inputs (irrelevant or incomplete frames). These pairs, after being refined by an LLM-based post-filtering step, are used for Direct Preference Optimization (DPO). While the paper reports performance gains on benchmarks like Video-MME and LongVideoBench, the novelty is limited, and the experimental rigor is insufficient to justify acceptance.

**Additional Comments:**

None

**Audience:**

Yes

**Audience Explanation:**

This paper focuses on an important question.

**Claims And Evidence:**

No

**Claims Explanation:**

**1. Flaws in Experimental Comparison Marginal Gains over SFT**
In Table 2, TPO shows only a 1.2% to 2.8% improvement over the "SFT Self" baseline (supervised fine-tuning on the same preferred data). Given that DPO is significantly more computationally expensive and requires complex hyperparameter tuning ($\beta$ and $\alpha$), these marginal gains do not convincingly demonstrate the superiority of the preference learning framework.

**2. Lack of Stronger Baselines**
The authors compare against "Hound-DPO," but they do not compare against more advanced preference optimization methods.

**3. Reliance on External Teacher Models**

(1) Cascading Errors: The pipeline depends heavily on CogVLM2 for captioning and GPT-4o-mini for question generation and post-filtering.

(2) Upper-Bound Constraint: The quality of the "preference" is fundamentally capped by the spatial-reasoning capabilities of these image-based models, rather than the video-LMM truly learning independent temporal dynamics from the video data itself.

**4. Incomplete Technical Analysis**

(1) Performance on Short Videos: The model shows comparable or negligible improvement on the Video-MME-short subset. This suggests that the method primarily helps the model ignore "noise" in long contexts rather than improving the precision of temporal localization.

(2) Computational Overhead: The paper lacks a detailed discussion on the trade-off between the training complexity of DPO (which requires keeping a reference model in memory) and the resulting performance boost.

**Requested Changes:**

Add results or explanation for the following questions.

**1. Flaws in Experimental Comparison Marginal Gains over SFT**
In Table 2, TPO shows only a 1.2% to 2.8% improvement over the "SFT Self" baseline (supervised fine-tuning on the same preferred data). Given that DPO is significantly more computationally expensive and requires complex hyperparameter tuning ($\beta$ and $\alpha$), these marginal gains do not convincingly demonstrate the superiority of the preference learning framework.

**2. Lack of Stronger Baselines**
The authors compare against "Hound-DPO," but they do not compare against more advanced preference optimization methods.

**3. Reliance on External Teacher Models**

(1) Cascading Errors: The pipeline depends heavily on CogVLM2 for captioning and GPT-4o-mini for question generation and post-filtering.

(2) Upper-Bound Constraint: The quality of the "preference" is fundamentally capped by the spatial-reasoning capabilities of these image-based models, rather than the video-LMM truly learning independent temporal dynamics from the video data itself.

**4. Incomplete Technical Analysis**

(1) Performance on Short Videos: The model shows comparable or negligible improvement on the Video-MME-short subset. This suggests that the method primarily helps the model ignore "noise" in long contexts rather than improving the precision of temporal localization.

(2) Computational Overhead: The paper lacks a detailed discussion on the trade-off between the training complexity of DPO (which requires keeping a reference model in memory) and the resulting performance boost.

---

### Note · Authors · 2026-04-01

I have read and agree with the venue's withdrawal policy on behalf of myself and my co-authors.